# Comparative Genomic Analysis Discloses Differential Distribution of Antibiotic Resistance Determinants between Worldwide Strains of the Emergent ST213 Genotype of *Salmonella* Typhimurium

**DOI:** 10.3390/antibiotics11070925

**Published:** 2022-07-09

**Authors:** Elda Araceli Hernández-Díaz, Ma. Soledad Vázquez-Garcidueñas, Andrea Monserrat Negrete-Paz, Gerardo Vázquez-Marrufo

**Affiliations:** 1Centro Multidisciplinario de Estudios en Biotecnología, Facultad de Medicina Veterinaria y Zootecnia, Universidad Michoacana de San Nicolás de Hidalgo, Km 9.5 Carretera Morelia-Zinapécuaro, Col. La Palma Tarímbaro, Morelia 58893, Michoacán, Mexico; 0617452b@umich.mx (E.A.H.-D.); andrea.negrete@umich.mx (A.M.N.-P.); 2División de Estudios de Posgrado, Facultad de Ciencias Médicas y Biológicas “Dr. Ignacio Chávez”, Universidad Michoacana de San Nicolás de Hidalgo, Ave. Rafael Carrillo esq. Dr. Salvador González Herrejón, Col. Cuauhtémoc, Morelia 58020, Michoacán, Mexico; soledad.vazquez@umich.mx

**Keywords:** ST213 worldwide strains, genomic databases, antibiotic resistance plasmids, Typhimurium

## Abstract

*Salmonella enterica* constitutes a global public health concern as one of the main etiological agents of human gastroenteritis. The Typhimurium serotype is frequently isolated from human, animal, food, and environmental samples, with its sequence type 19 (ST19) being the most widely distributed around the world as well as the founder genotype. The replacement of the ST19 genotype with the ST213 genotype that has multiple antibiotic resistance (MAR) in human and food samples was first observed in Mexico. The number of available genomes of ST213 strains in public databases indicates its fast worldwide dispersion, but its public health relevance is unknown. A comparative genomic analysis conducted as part of this research identified the presence of 44 genes, 34 plasmids, and five point mutations associated with antibiotic resistance, distributed across 220 genomes of ST213 strains, indicating the MAR phenotype. In general, the grouping pattern in correspondence to the presence/absence of genes/plasmids that confer antibiotic resistance cluster the genomes according to the geographical origin where the strain was isolated. Genetic determinants of antibiotic resistance group the genomes of North America (Canada, Mexico, USA) strains, and suggest a dispersion route to reach the United Kingdom and, from there, the rest of Europe, then Asia and Oceania. The results obtained here highlight the worldwide public health relevance of the ST213 genotype, which contains a great diversity of genetic elements associated with MAR.

## 1. Introduction

*Salmonella enterica* is one of the main pathogens associated with food contamination; it is considered responsible for around 94 million cases of gastrointestinal illnesses and 155,000 annual deaths worldwide [1,2,3]. The typing method used for the follow-up of outbreaks and epidemiological studies of *S. enterica* for nearly 90 years is serotyping [4]. Currently, more than 2600 serotypes are registered worldwide [5,6]. The Typhimurium and Enteritidis serotypes are considered of the greatest global public health relevance, because these have the widest geographical distribution and the highest incidence in clinical and food samples worldwide [7,8]. However, among the plethora of molecular genetic typing methods generated in the last few decades, comparative genomic analysis stands out for its greater discrimination power, making it possible to distinguish strains associated with an outbreak from those that are not [9,10,11]. This discrimination power is of epidemiological and public health relevance, since it allows the generation of strategies for the prevention and control of outbreaks [12,13,14].

Among the genotyping methods used as epidemiological tools for the study of *S. enterica*, multi locus sequence typing (MLST) for the assignment of sequence type (ST) through variations in seven loci efficiently identify clonal groups and founder genotypes [15,16]. Recently, MLST analysis was modified in accordance with the possibilities offered by whole-genome sequencing (WGS) to use a large number of genes from the core genome [17,18,19], enabling the differentiation between clonal groups to be more precise, allowing the description of emerging genotypes, and making it possible to distinguish between lineages within an ST [20,21,22]. This resolution power offered by MLST using WGS opens the possibility of comparing strains between very diverse space-time scales; in addition to its application in epidemiological studies, this allows evaluating the micro-evolutionary process of *S. enterica* and detecting the emergence of variants of relevance in the field of public health [21,22,23]. Beyond its excellent ability to discriminate between clonal groups and variants within an ST, comparative genomic analysis allows the study of dispersion and distribution of virulence and resistance to antibiotics-associated genes [24,25,26,27,28] and relevant genetic determinants of *S. enterica* emerging pathogenic variants.

MLST analysis has clearly established that the founding genotype of *S. enterica* Typhimurium is ST19, as the most prevalent genotype of this serotype around the world from which the vast majority of other STs have been derived [15,16]. However, it has been recently documented that some countries have experienced the ST19 being replaced with other STs that show a higher incidence in clinical and food samples. These replacement genotypes have genotypic and phenotypic characteristics that make them relevant in terms of epidemiology and public health. In this sense, the best documented cases of replacement are replacements by the ST313 genotype, which have been identified in Sub-Saharan Africa and are associated with systemic disease in HIV patients. Thus far, two sub-lineages have been identified, one of which has resistance to antibiotics, complicating the treatment of HIV [21,29]. The increased incidence of the ST34 genotype carrying antibiotic resistance genes/plasmids has been documented in some regions of China in both clinical and food samples [30,31].

In Mexico, a study on clinical and food samples carried out for four years in different states of the country revealed evidence of the replacement of ST19 by ST213 [32]. Subsequent analyses showed that this strain carries IncA/C plasmids, now changed to IncC [33], with genetic determinants for multiple resistance to antibiotics [34]. According to the genome metadata available in EnteroBase [35], the ST213 genotype has been isolated in recent years from different regions of the world. However, to date the epidemiological risk associated with the geographical dispersion of this genotype sequence is unknown, as is whether the patterns of resistance to antibiotics are shared between strains from different regions.

It has been documented that bioinformatic genomic analysis allows the robust prediction of phenotypic resistance to antibiotics in *S. enterica* [28,36]. Thanks to the considerable increase in the available genomes of ST213 from different regions of the world, it is feasible to perform an analysis that allows establishing solid hypotheses about the resistance of this genotype to antibiotics. Furthermore, it is possible to establish whether the strains from different geographical regions have the same set of genetic determinants of resistance to antibiotics and the presence of multi-resistance between them. Therefore, the objective of this work was to carry out a comparative genomic analysis of all ST213 strains whose genomes are available in public databases in order to analyze the presence and distribution of genes and mutations associated with antibiotic resistance. Differences in the presence/absence of genetic determinants associated with resistance to antibiotics in relation to the geographical origin, year of isolation, and type of samples of ST213 genotype strains are analyzed and discussed.

## 2. Results

### 2.1. Distribution of the Genomes of the Strains Analyzed by Country, Type of Sample and Year of Isolation

Of the 220 genomes retrieved from databases of S. Typhimurium strains belonging to the ST213 genotype and included in this study, 29% (*n* = 64) came from the United Kingdom, 25.9% (*n* = 57) came from Mexico, 24.5% (*n* = 54) came from the United States of America, and 9.5% (*n* = 21) came from Canada, while the remaining 11% (*n* = 24) were obtained from nine different countries, mainly from Europe (Figure 1). 

A total of 70% (*n* = 154) of the strains were obtained from human clinical samples, 15% (*n* = 33) were obtained from samples of animal origin, and the remaining 15% (*n* = 33) were obtained from nine other sources in smaller percentages (Figure 2). In relation to the year of isolation, the genomes were derived from strains isolated between the years 1957 and 2021, with the years 2003, 2004, 2017, 2018, and 2019 being the ones in which the largest number of records were made. Between 2002 and 2005, most of the Mexican records were made; between 2014 and 2019, the largest number of records from the United Kingdom were obtained; and the strains from the United States were all registered in 19 years, apart from the years 1965, 1967, 2001, 2004, 2005, 2008, and 2009. It is interesting to note that, so far, during the year 2013, no isolates of ST213 have been recorded in any geographical area (Figure 3).

### 2.2. Presence of Antibiotic Resistance Genes

Forty-four antibiotic resistance genes were identified in the analyzed genomes (Table 1), of which *aac(6)-Iaa*, *golS*, *mdsA*, *mdsB*, *mdsC*, *mdtK*, and *sdiA* are present in the 220 (100%) genomes analyzed. The genes *aac(3)-IV*, *aph(3)-IIa*, *aph(4)-Ia*, *bla_CARB-3_*, *catII*, *cmlA1*, *dfrA1*, *linG*, *mefB*, *qnrA1*, *qnrB19*, *qnrS1*, *tetM*, and *tetU* were found in <1% of the genomes studied. *The aadA5*, *ant(3)-IIa*, *aph(3)-Ia*, *bla_TEM-1_*, *dfrA17*, *qnrB5*, *sul1*, *tetB*, and *tetC* genes were found in between 1 and 15% of the genomes included in this study, while the *aac(3)-IId*, *aadA2*, *bla_CMY-59_*, *dfrA12*, *floR*, *oqxA*, *oqxB*, *qacH*, *sul2*, and *sul3* genes were identified in 20% to 40% of the study genomes. Finally, the *aph(3)-Ib*, *aph(6)-Id*, *tetA*, and *tetR* genes were found in 50–57% of all the analyzed genomes (Table 1). Regarding the type of sample from which the genomes were isolated, the *bla_CARB-3_* gene was only found in strains from samples of animal origin, the *bla_CMY-59_* gene was not found in strains from food, and *bla_TEM-1_* was identified in strains taken from water samples. On the other hand, the *linG* gene was found in food-associated strains and *mefB* was identified in human clinical samples from the United Kingdom and the United States, respectively. Regarding the country from which the strains were obtained, ten of the genes that confer resistance to aminoglycosides were found in the genomes of strains from Australia, Canada, Denmark, Thailand, Mexico, the United Kingdom, and the United States. Genes related to resistance to β-lactams and fluoroquinolones were not found in the genomes of strains from Denmark, and genes related to resistance to chloramphenicol were not found in the genomes of strains from Thailand. Similarly, tetracycline and diaminopyrimidine resistance genes were not found in strains from Denmark and Thailand, while the *qacH* gene was not identified in the genomes of strains isolated in Australia.

### 2.3. Antibiotic Resistance Mutations

A total of five point mutations associated with codon/amino acid changes that confer resistance to fluoroquinolones were found in the analyzed genomes, four of which were found in the *gyrA* gene and one of which was found in the *parC* gene (Table 2). The most frequent mutation was p.S83Y, which was carried by 23 genomes, whereas the least frequent were p.D87G and p.S80I, which were present in one genome each. Except for the genome of one Thailand strain, the rest of the genomes carrying such mutations belonged to North American strains, with one coming from Canada, nine from Mexico, and 19 from the USA (Table 2).

### 2.4. Plasmid Replicons Detection

Thirty-three different plasmid replicons were found in the analyzed genomes, of which 23 were identified in fewer than 5% of the study genomes, seven in more than 5% but fewer than 30%, two in 34%, and one in 47.7%. However, twelve of the analyzed genomes did not present plasmid replicons. Of the detected plasmids, it was recently proposed that the IncA/C2 denomination must be discarded and replaced by the IncC nomenclature [33]; thus, in the present paper, all bioinformatic detection of IncA/C2 in this work was designed as IncC. Despite the bioinformatics detection of plasmid replicons, it must be taken into account that, although this occurs infrequently, plasmids can be integrated in the bacterial chromosome or co-integrated in a plasmid with multiple replicons [37,38].

In relation to the type of sample used, the ColRNAI, IncC, and IncFIB(K)_1_Kpn3 plasmids were present in all the genomes of all the sample types analyzed here. Plasmids IncP1 and IncQ1 were only found in strains of samples of animal origin, while plasmids Col(BS512), IncFIC(FII), IncFII(pCTU2)_1_pCTU2, IncI2, IncI2_1_δ, IncX4, pESA2, pSL483, and rep14a_4_rep(AUS0004p3) were identified only in strains obtained from human clinical samples. In the case of the plasmids Col(MG828), Col440II, IncFIA, IncFIB(AP001918), IncFII, IncFII(pCoo)_1_pCoo, and IncX, these were found in genomes whose strains came from samples of both animal and human origin. Plasmids IncHI1A and IncHI1B(R27)_1_R27 were identified in strains from samples of animal and undefined origin, while Col156 and ColpVC were found in strains of animal and human origin as well as in food samples. Plasmid IncFIA(HI1)_1_HI1 was only found in strains from undefined samples; IncI_γ_1 was found in strains of animal, human, and undefined origin; and Col440I was not found in the genomes of food-associated strains. Plasmids IncFIB(S), IncFII(S), IncHI2A, IncHI2, and IncI1_1_α_1 were not identified in strains identified from water samples. On the other hand, the plasmids IncFIC(FII), IncI2_1_δ, pESA2 rep14a_4_rep(AUS0004p3), and IncFII(pCTU2)_1_pCTU2 were identified in the genomes of strains isolated in Mexico; pSL483 was identified only in strains from Canada; IncI2 was identified only in strains from the United Kingdom, and Col( BS512), IncFIA(HI1)_1_HI1, IncFIA, IncP1, and IncQ1 were identified only in strains from the United States. Plasmids IncX4 and IncI_γ_1 were found only in the genomes of strains from the United States and the United Kingdom; IncFII was found only in strains from Canada and the United States; and IncFIB(AP001918), IncFII(pCoo)_1_pCoo, IncHI1A, IncHI1B(R27)_1_R27, IncHI2, and IncHI2A were identified only in strains from Mexico and the United States. In the case of the Col(MG828) and IncX plasmids, they were identified in strains from Canada, Mexico, and the United States; Col156 and Col440II in strains from Canada, the United States, and the United Kingdom; ColpVC were identified in strains from Mexico, the United States, and the United Kingdom; Col440I and IncFIB(K)_1_Kpn3 were identified in strains from these three North American countries and the United Kingdom; and ColRNAI and IncC were identified in strains from North America, the UK, and Australia. Finally, the IncI1_1_α_1 plasmid was not found in strains from Australia, France, India, Ireland, the Netherlands, or Portugal, and the IncFIB(S), IncFII(S) plasmids were not present in the genomes of strains from Mexico and Portugal.

### 2.5. Grouping Patterns

When the presence/absence of resistance genes in the study genomes is used as a matrix for the generation of clustering patterns, interesting groups can be observed in relation to the country and the sample of origin of the strains. The presence of resistance genes brings together the strains from the three North American countries (Canada, Mexico, USA), as well as those from the United Kingdom and Australia. Mexico and the United States appear together in a subcluster while Canada, the United Kingdom, and Australia are grouped in another branch (Figure 4). In the other large cluster, the strains from European countries appear alongside strains from India and Thailand, with the strains from the latter and Denmark being distinguished by the presence of particular resistance genes that have previously been mentioned. Regarding the clustering pattern of the heat map generated when information on the presence/absence of resistance genes in the analyzed genomes is combined with the type of sample used, the genomes from chicken-, water-, swine-, and non-determined-origin strains cluster in the first group (Figure 5). Food and bovine samples are grouped together, and human samples are segregated in a single terminal ramification. Genomes derived from poultry and canine strains cluster in the last group. 

Regarding the grouping pattern created by the presence/absence of resistance genes from each individual genome/strain, it can be seen that two larger clusters are clearly defined (Figure 6). The first includes mainly genomes from UK strains along with two Canadian genomes and all the European, Asian, and Australian samples (Figure 6, cluster A). This first cluster also includes, though in a different subgroup, several North American samples (Canada, Mexico, USA), two samples from the UK, and one sample from Thailand. The second main cluster contains genomes/strains exclusively from the three North American countries, and only three UK samples are dispersed in the three subgroups of this cluster (Figure 6, cluster B).

The plasmid replicons presence/absence grouping pattern shows a similar pattern to that observed for previously described resistance genes. Genomes of countries from North American (Canada, Mexico, USA) strains shape the first big cluster, with one strain from the UK and one from Australia (Figure 7, cluster A). The second big cluster is composed of European, Asian, and Australian genomes/strains, with five samples from Canada, one from Mexico, and one from the USA dispersed in the subgroups of the cluster (Figure 7, cluster B). A total of three small clusters are made up of genomes of strains from Mexico and the USA, as well as one from the UK (Figure 7, clusters C–E).

## 3. Discussion

In the present work, the intragenotypic variability of *S*. Typhimurium ST213 strains in relation to the presence of genes associated with antibiotic resistance was evaluated by means of comparative genomic analysis. In the same way, this variation was documented in relation to the country and the type of sample of origin of the strains whose genomes were analyzed. Previous analyses related to the distribution of *S. enterica* strains in different countries and sample types, as well as the variability in the distribution of genetic determinants of antibiotic resistance, have focused mainly on evaluating the differences between serotypes or within the same serotype, as shown by several recent reviews on the subject [2,8,24,39,40]. Epidemiological studies documenting genomic variation associated with antibiotic resistance at a more subtle level, such as within the same *S. enterica* ST genotype, despite having been performed increasingly more frequently in recent years [29,41,42,43], are still relatively scarce. In this sense, the description of the variation in the resistance genes of *S.* Typhimurium strains of the ST313 genotype that causes systemic infection in Africa stands out, with differences having been observed in the phenotypic pattern of resistance to antibiotics associated with both the presence of the plasmid named pSLT-BT as well as with the composition of genes within it [21,29]. Similarly, strains of the emerging genotype ST34 present different patterns of resistance to antibiotics and show variation in the presence of genetic determinants of resistance [31,43]. As far as we have been able to document, these are the cases in which intragenotypic differences have been established in the epidemiology and patterns of antibiotic resistance in *S. enterica* in emergent/re-emergent STs replacing the ancestral ST19 genotype.

The emerging ST213 genotype was reported in Mexico to be associated with a process of displacement of the ST19 genotype [44], considered to be the founder within the Typhimurium serotype. The process of genome retrieval from public databases of strains of the ST213 genotype carried out here shows that this genotype is now frequently isolated in North America (Canada, Mexico, USA) and has recently spread to Europe, Asia, and Oceania, which is why it may represent a global public health problem as an emerging/re-emerging genotype.

It is noteworthy that six of the ST213 genomes retrieved from the databases belong to stored strains isolated before this century, with the oldest one in the USA dating back to 1957. This finding clearly shows that ST213 genotype was present long before its detection as an emerging public health concern in Mexico, replacing the founder ST19 genotype [44]. Additionally, the genomes retrieved indicate that ST213 strains have been widely dispersed throughout the North America region (Canada, Mexico, USA) since the beginning of this century. Furthermore, despite its presence in Europe (specifically Portugal) since 1965, strains of the ST213 genotype began to be frequently recovered from clinical and food samples at the beginning of this century, particularly in the United Kingdom, but also reached Thailand at the same time. The isolation year of ST213 strains whose genomes were analyzed here suggests its recent dispersion to India and Australia. 

The fast spread of strains of the ST213 genotype in North America makes sense, given the geographical proximity and the constant flow of people and food between the three countries of this region [45,46,47]. Despite the need for a detailed epidemiological analysis, the years of isolation of the ST213 strains analyzed here suggest a dispersion route since its emergence and detection in Mexico. The available data strongly suggest that despite their global presence of at least 65 years, the ST213 strains have only recently become a public health issue because of its emergence in Mexico and/or the USA. The grouping presence/absence patterns of plasmids and resistance genes indicate that ST213 reached the UK from the USA and, from this country, spread to the rest of Europe, Asia, and Oceania. It is plausible that the ST213 genotype dispersed from the USA to the UK and, from there, to the rest of Europe, Asia, and Oceania through a traveler, rather than food or a vehicle. The traveler income/outcome of multi-resistant *S. enterica* strains of different serotypes has recently been documented in the regions and countries involved in the present analysis [48,49,50,51,52]. As previously stated, this epidemiological pattern is a hypothesis that emerged from the results obtained here and deserves further genomic epidemiological analysis.

The grouping pattern generated based on the sample of isolated strains whose genomes were analyzed here indicates that human strains possess a particular set of antibiotic resistance genetic determinants. However, the determinant genetics of antibiotic resistance from human samples share a bigger clade with animal and meat (food) samples. Thus, such a grouping pattern suggests that ST213 human strains are mainly acquired from farm animals and food and/or interchange some genetic subset(s) with strains from these sources. This hypothesis is consistent with reports of the frequent zoonotic transmission of non-invasive Typhimurium serotype strains in Mexico [32,53], Canada [54,55], and the United States [56,57], the three North American countries, which contributed the greatest number of strains to this study. Furthermore, in the zoonotic reports from these three countries, the transmission of strains of *S. enterica* from pigs, cattle, and poultry to humans is common, and these are the species samples whose genomes were analyzed in this work. 

All antibiotic resistance genes detected in the ST213 strains genomes analyzed were predominantly relevant in the North American countries. In the same way, all the resistance genes/mutations were present predominantly in human samples. Both regional and sample type predominancies can be partially explained by the overrepresentation of genomes from strains at regional and sample levels. Furthermore, Mexico or North America are the probable origin of the ST213 as an emergent/reemergent pathogen of global health relevance. Thus, the discussion highlights results for North America and the human ST213 genomes. The ST213 genomes harbor a great diversity of genetic determinants for antibiotic resistance, and the gene/mutations/plasmid replicon combinations per genome are also diverse. The genetic determinants to resistance against quinolones and aminoglycosides are relevant because the World Health Organization classifies them as critically important antimicrobials which are the “Sole, or one of limited available therapies, to treat serious bacterial infections in people”, along with other relevant criteria [58]. 

The *qnr* and *oqxAB* genes associated with quinolone resistance are mainly related to plasmids [59,60]. Despite being found in other serotypes in Mexico, the *qnr* genes were absent in Typhimurium strains of animal origin [61,62], but present in Mexican human strains of this serotype [63]. The *oqxAB* genes found here have also been previously reported for several serotypes of human isolates in Mexico [61,62]. Resistance to quinolones is considered low for *S. enterica* strains in the USA, although in this country it has been documented the presence of genes conferring resistance to such antimicrobial compounds in the serotype Typhimurium strains coming from clinical, pork meat, and livestock samples [64,65,66,67]. In contrast, in the case of Canada, the *qnrB* and *qnrS* genes were recently reported in ciprofloxacin-resistant human clinical strains of different serotypes of *S. enterica*, including Typhimurium, and the *oqxAB* gene has rarely been found [68]. Regarding the five point mutations that occur in the quinolone resistance-determining regions (QRDR) of DNA gyrase (*gyrA*) and topoisomerase IV (*parC*) [60] found in the genomes of ST213 strains, the double mutant S83Y in *gyrA* and S80I in *parC* is interesting, since *S.* Typhimurium mutations in *gyrA*, in addition to playing a dominant role in resistance to fluoroquinolones, have a synergistic effect with other resistance mechanisms, while the S80I mutation in *parC* appears to have no effect on quinolone resistance without *gyrA* mutations [69]. Whereas QRDR mutations are frequent in bovine *S. enterica* isolated strains of different serotypes in Mexico [62] and Canada [70], it appears to be absent or to occur with low frequency in the USA [36,71].

In relation to aminoglycoside resistance, the ST213 genomes carry the genes coding for the three types of modifying enzymes [72]. However, the CLSI has cautioned that aminoglycosides are not clinically effective against *S. enterica*, with this species being relevant as a carrier and potentially a means of transfer to other pathogenic bacteria [73]. The transference of genomic island one of *S. enterica* to *Escherichia coli*, a mobilizable element that carries multiple resistance antibiotics genes, has been experimentally demonstrated [74]. It has been documented that gentamicin resistance has increased significantly in the strains of eight serovars in human and retail meat in Canada in the last few decades [75]. Around 50% of Typhimurium strains from different sources from Mexico now show gentamicin resistance [32] Additionally, aminoglycoside resistance genes are highly prevalent in *S. enterica* strains isolated from farm animals in the USA, showing the geographical variations in its incidence [76].

In the case of tetracycline, sulfonamides, and phenicols, these antibiotics are considered by the WHO in the category of highly important antimicrobials, fulfilling the same first criterion of the previous antimicrobials but not adding more criteria [58]. The six tetracycline resistance genes detected in the ST213 strains are related to two resistance mechanisms, efflux pumps and ribosomal protection [77]. Regarding the sulfonamide resistance genes, the ST213 genome carries three of the four resistance genes associated with dihydropteroate synthetase modifications [78]. Interestingly, *sul2* and *sul3* predominate over the most frequently reported *sul1*. Additionally, the genomes carry three modifications of the dihydrofolate reductase enzyme for resistance to trimethoprim, of which *dfrA12* predominates. Although three genes that confer resistance to phenicols in the ST213 genomes are also present, the most widely distributed is the efflux bomb-coding gene *floR*. The strains of the Typhimurium serotype isolated from different sources show high percentages of resistance to these three types of antibiotics [32].

Besides all the previously commented resistance genes, all the analyzed genomes harbor *aac(6′)-Iaa*, *mdsA*, *mdsB*, *mdsC*, *golS*, *sdiA*, and the majority carried the *qacH* gene. The *aac(6’)-Iaa* gene for aminoglycosides resistance resides in the chromosome of Typhimurium strains and is present in the ancestral ST19 genotype [79]. The genes *mdsA*, *mdsB*, and *mdsC* code for the transporter efflux pump mdsABC *golS*, a promoter that is related to resistance to novobiocin [80], And it is also present in the ST19 genotype [81]. The *sdiA* gene codes for a quorum-sensing regulator that mediates the multi-drug resistance AcrAB efflux pump [82], the overproduction of which confers multidrug resistance; it is also present in the ST19 genotype. Finally, the *qacH* gene codes for an efflux bomb that confers resistance to the quaternary ammonium compounds, commonly used organic disinfectants [83]. This last gene is mainly found in different types of mobile genetic elements and confers resistance to aminoglycosides, chloramphenicol, erythromycin, and tetracyclines. As a whole, these antecedents suggest that such common ST213 genes are ancestral characters present in the ST19 genotype that can confer antibiotic multi-resistance to both genotypes.

All the antibiotics mentioned above are relevant for their use in veterinary and human medicine [58,84]. The samples of isolation of the ST213 strains for which genomes were analyzed here were predominantly from human and farm animals. This suggests that both farms and hospitals are relevant reservoirs for *S. enterica* ST213 strains carrying resistance genes that can be transferred to other pathogenic bacteria or directly to humans [85]. The diversity and distribution of the genes/mutations found in the analyzed genomes suggest that most of the North American ST213 strains are phenotypically multi-resistant to antibiotics.

In this work, 33 different plasmids were bioinformatically identified in the genomes of ST213 strains from around the world. Inc-type plasmids were found in 207 strains, while Col-type plasmids in 103; pESA2, pSL483 and rep14a_4_rep were found in a single strain each. The genomes of *S. enterica* North American ST123 strains from the US, Mexico, and Canada harbor 32 of the 33 plasmids, in contrast to the reduced variation in genomes of strains from Europe, but the low number of ST213 strains from Asia and Australia makes comparison difficult within these regions.

The Inc (C, F, H, I1) conjugative plasmids found in the analyzed genomes are frequently reported in different *S. enterica* serotypes, including Typhimurium, which together carry genes for all kind of antibiotics, some of which can be within an integron [37,38]. The simultaneous presence of IncHI2/IncHI2A and IncFIB(S)/IncFII(S) plasmids in the same ST213 genomes has previously been observed in *S. enterica* strains from Asia [86], Europe [87], and South America [88] for the first pair and in Africa for the second [89]. In addition, the IncHI2/IncHI2A pair was only found in the genomes of North American strains, while the presence of the IncFIB(S)/IncFII(S) pair was predominant in European strains. The plasmids IncI2 [90], ColpVC, IncHI2/IncHI2A, IncFIA, IncH1A, IncFIA(HI1)_1_HI1 [88,91], and ColRNAI, IncFIB(S)/IncFII(S) [41,92] have already been reported in strains of the ST19 founder genotype. The Col and IncX plasmids were present but less frequent in the ST213 genomes analyzed. The Col plasmids are mainly associated with the spread of *qnrS1* and *qnrB19* [37], and specifically the ColpVC detected here is associated with *aph(3″)-Ia* in bovine strains [93]. The IncX plasmids have been mainly isolated from human and animal samples and mainly encode genetic determinants against extended-spectrum β-lactams and quinolones [37,38]. 

The presence of the replicon rep14a_4_rep(AUS0004p3) is surprising because this plasmid belongs to a three-membered family of small mobilizable plasmids only described in the Gram-positive species *Enterococcus faecium*; thus, it must be considered to have a narrow host range [94]. Furthermore, the presence of these plasmids in *E. faecium* is associated with tetracycline resistance in strains isolated from human clinical samples, and it has been stated that such plasmids cannot be transferred by conjugation. This intriguing finding indicates the existence of a horizontal gene transfer (HGT) mechanism between *E. faecium* and *S. enterica*, which, to the best of our knowledge, has not previously been suggested. Further data on the plasmid sequence are needed to corroborate this result, and it will be relevant to analyze the HGT mechanism between the involved species.

The diversity of plasmid replicons found in the ST213 genomes indicates that such genotypes can be prone to the horizontal gene transfer (HGT) of antibiotic resistance genes. The plasmid diversity and distribution in the genomes analyzed here indicates that the plasmid interchange in ST213 strains features different dynamics between North America and Europe, apparently being more dynamic in the former region. Several plasmid replicons were found in the genomes of ST213 strains from all types of samples but showed different rates of incidence. However, most plasmids are present in samples related to the well-established zoonotic/food transmission chains of *S. enterica* to humans. Further work is needed to analyze a possible positive relationship between the type of sample and the presence/absence of a particular type of plasmid. Additionally, it is important that further studies define whether there is any significant relationship between the presence of these plasmids and the geographical region of origin of the ST213 strains, as well as its public health consequences.

It is important to consider that one of the weaknesses of the present study is that the analyzed genomes did not come from a systematic program based on the random sampling of strains. All ST213 genotype genomes available in public databases, of good quality, and with adequate metadata were analyzed. Thus, the trends and rates of prevalence reported here are the result of various factors, including the study of specific outbreaks by country as well as the capacity for genomic sequencing and its use as an epidemiological surveillance tool in different countries. The effect of these variables is reflected in the number of genomes of the ST213 genotype reported by year and by country (Figure 3), where the prevalence of genomes of strains from the USA, the UK, and other European countries, as well as from Australia, can be observed. Some of these countries routinely use genome sequencing in outbreak analyses and have adopted such strategies as the standard for the epidemiological surveillance of *S. enterica* or are in the process of doing so [95,96,97,98,99].

Another factor that may influence the results of the analysis carried out here is the quality of the sequences used, since low-level contamination can affect the detection of antibiotic resistance genes and point mutations associated with resistance [100]. However, the assembled genomes of EnteroBase undergo a quality control process that excludes the effects of such contamination [35], and the sets of reads used were filtered with bioinformatic tools to remove those that did not meet appropriate quality standards. Furthermore, it has been documented that the bioinformatic analysis of WGS to predict antibiotic resistance in *S. enterica* can generate results inconsistent with the observed phenotypic resistance [101]. However, in such works, only the ResFinder 3.2.0 tool is used for the location of antibiotic resistance genes, in contrast to the use of this software together with the ARG-ANNOT and CARD tools here to aid in locating the genetic determinants of antibiotic resistance, along with the use of databases recently curated to include genes and mutations previously not included in ResFinder [96,97,98]. In addition, the ResFinder update makes it possible to determine the number of copies of genes associated with resistance [102], another factor in antibiotic resistance that has previously not been considered by bioinformatics packages in resistance determination. Despite the predictive power of this bioinformatic analysis, it was desirable to contrast the results obtained here with the phenotypic antibiotic resistance/susceptibility data in order to obtain a clearer picture of the public health relevance of the ST213 genotype.

## 4. Materials and Methods

### 4.1. Data Retrieval

A total of 286 sets (572 reads pairs clusters) of Illumina sequencing reads available in EMBL’s European Bioinformatics Institute (https://www.ebi.ac.uk/, accessed on 1 July 2022) and 33 EnteroBase assembled genomes (https://enterobase.warwick.ac.uk/, accessed on 1 July 2022) [35,103,104] assigned to ST213 genotype strains of *Salmonella enterica* were retrieved. All read sets and assembled genomes retrieved were filtered for a quality assessment to exclude duplicates, remove low-quality reads, and sampling coverage (Figure 8). Additionally, reads/genomes for which relevant metadata (country of isolation, year of isolation, sample kind) were not available were discarded. After filtering (see Section 4.2 below), 188 read sets and 32 assembled genomes were deemed to be adequate for analysis (Appendix A).

### 4.2. Bioinformatic Analysis

The 188 sets of sequencing reads and 32 assembled genomes were processed with TORMES 3.0 [105] for sequence quality filtering, de novo genome assembly, and antibiotic resistance gene screening. All the genomes used in this analysis were at the level of *draft* genomes. The quality parameters used in the bioinformatic filtering to select both the genomes obtained from Enterobase (Appendix A) and for the genomes assembled in this work (Appendix A) were a high value of N50, an average length of contigs of greater than 5000 bases, and a low number of *contigs* [106], Additionally, the genome length obtained was in agreement with the genome length of different serotypes of *Salmonella enterica*, including Typhimurium [107]. Regarding the number of contigs, the mean of the assemblies obtained from the Enterobase was 73, while the mean for the assemblies carried out in this work was 89. This allowed us to ensure an adequate quality of the assemblies used for the gene search analysis and identification.

The identification of antibiotic resistance genes in the genomes of interest was carried out using ARG-ANNOT [108], CARD [109,110], and ResFinder 3.2.0 [111]. The PlasmidFinder tool [112] was used for the identification of plasmid replicons and PointFinder 3.1.0 for mutations in antibiotic resistance genes [113]. Gene nomenclature used here is provided by CARD.

### 4.3. Data Analysis

Binary (1,0) matrices were constructed to represent the presence (1)/absence (0) of resistance genes or plasmids, as appropriate. The genes/plasmids present in the genomes of all tested strains were excluded from this analysis. Both matrices were analyzed in the Heatmapper server [114] with the parameters of the Manhattan and UPGMA criteria for the calculation of distances and the generation of the grouping pattern, respectively. Manhattan distances were calculated based on the presence/absence of 37 identified antibiotic resistance genes, excluding seven genes that were present in 100% of the analyzed genomes (see Table 1). Additionally, those strains lacking all the genes/plasmids were excluded from this analysis.

## 5. Conclusions

The replacement of the ST19 genotype by other genotypes in different countries represents a serious public health concern worldwide because of the virulence and multiple antibiotic resistance of these emergent genotypes. Thus, the fast increase in the number of genomes deposited in public databases belonging to strains of the ST213 genotype throughout the world is a signal of the need to study this multiple-antibiotic-resistant and virulent *S.* Typhimurium variant. The great number of antibiotic resistance genes and plasmid replicons carried by ST213 strains and its simultaneous presence in the same genome indicates that most of them present a multiple antibiotic resistance phenotype and are prone to the HGT of antibiotic resistance determinants. Additionally, the genomes and grouping patterns obtained here suggest a route of dispersion of the ST213 emergent genotype beginning in North America (Canada, Mexico, USA) and moving to the United Kingdom, with further dispersion occurring from here to the rest of Europe and simultaneously to Asia and Oceania. Further detailed epidemiological analysis is necessary in order to clarify this dispersion hypothesis and to understand the mechanisms associated with the differences in the pattern of the presence/absence of antimicrobial resistance genetic determinants by the geographical origin of the strains and the type of sample of precedence.

## Figures and Tables

**Figure 1 antibiotics-11-00925-f001:**
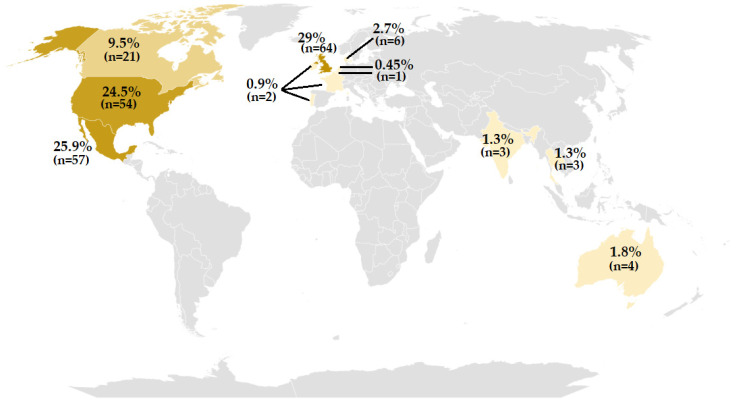
World distribution of the *Salmonella enterica* Typhimurium strain genotype ST213, the genomes of which were analyzed in this work.

**Figure 2 antibiotics-11-00925-f002:**
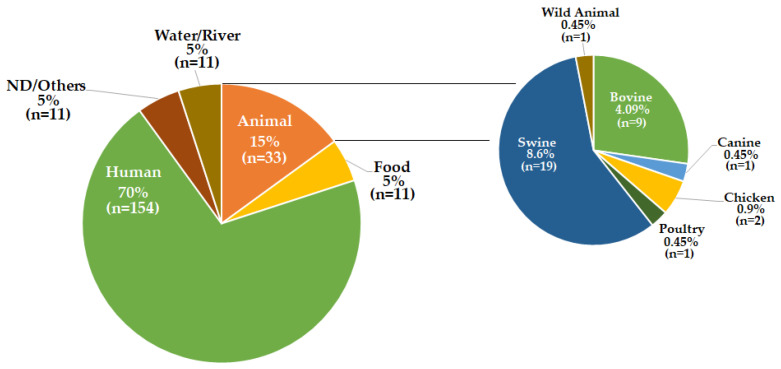
Sample type provenance of the ST213 strains from which genomes were analyzed in this work. The number of strains (*n*) for each sample type is given in parenthesis. The total number of genomes analyzed after filtering was 220 (see Figure 8).

**Figure 3 antibiotics-11-00925-f003:**
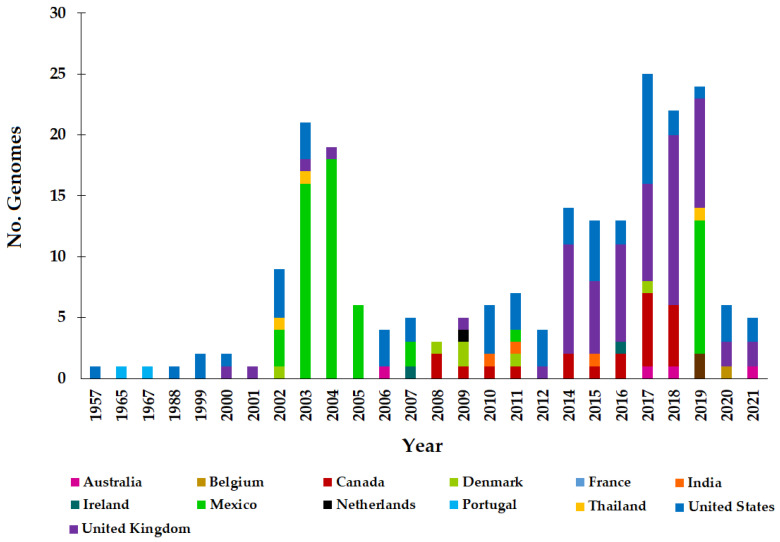
Year of isolation of the ST213 strains from which genomes were analyzed in this work. The total number of genomes analyzed after filtering was 220 (see Figure 8).

**Figure 4 antibiotics-11-00925-f004:**
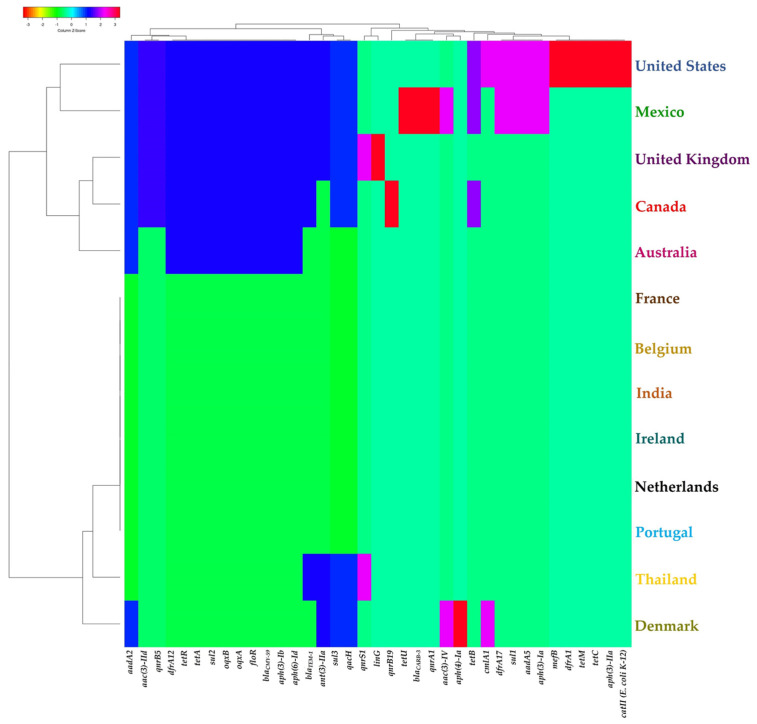
Grouping pattern by country of isolation and resistance genes identified in the genomes of the ST213 strains analyzed in this work. A binary matrix of presence (1)/absence (0) of each gene shown was used as an input for grouping using Manhattan and UPGMA for the calculation of distances and the generation of the grouping pattern, respectively. See the Section 4 for details.

**Figure 5 antibiotics-11-00925-f005:**
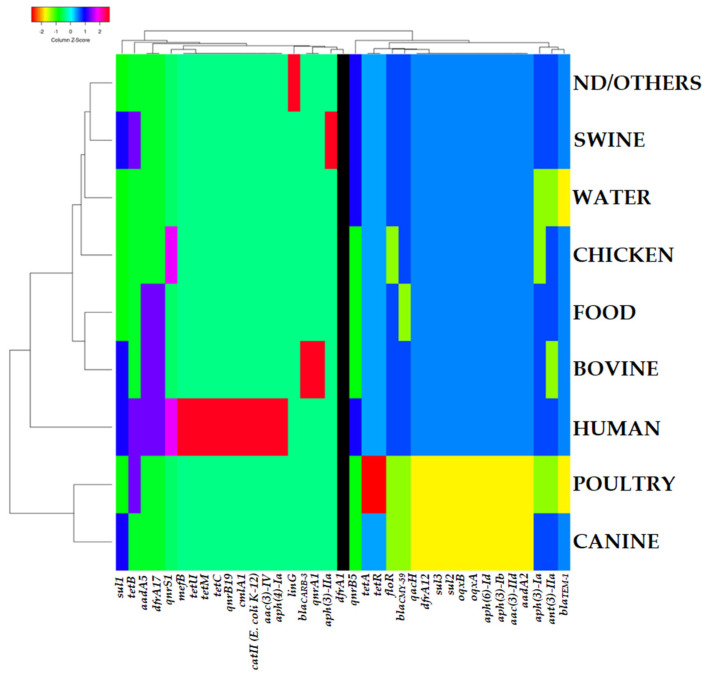
Grouping pattern by type of sample of isolation and resistance gene identified in the genomes of the ST213 strains analyzed in this work. A binary matrix of the presence (1)/absence (0) of each shown gene was used as an input for grouping using the Manhattan and UPGMA methods for the calculation of distances and the generation of the grouping pattern, respectively. See Section 4 for details.

**Figure 6 antibiotics-11-00925-f006:**
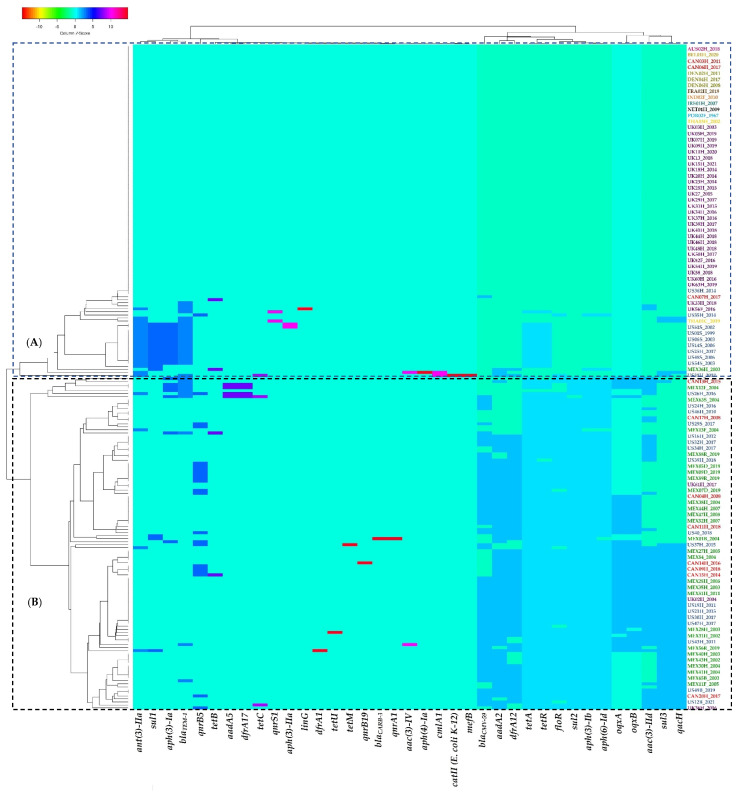
Grouping pattern by strain and resistance gene identified in the genomes of the ST213 strains analyzed in this work. A binary matrix of presence (1)/absence (0) of each shown gene was used as an input for grouping using the Manhattan and UPGMA methods for the calculation of distances and the generation of the grouping pattern, respectively. See Section 4 for details. (**A**) Grouping pattern of genome/strain from UK strains along with two Canadian ge-nomes and all the European, Asian, and Australian samples; (**B**) Grouping pattern of ge-nome/strain from the three North American countries, and only three UK. Country symbols: **AUS**, Australia; **BEL**, Belgium; **CAN**, Canada; **DEN**, Denmark; **FRA**, France; **IND**, India; **IRE**, Ireland; **MEX**, Mexico; **NET**, Netherlands; **POR**, Portugal; **THA**, Thailand; **UK**, United Kingdom; **US**, United States of America.

**Figure 7 antibiotics-11-00925-f007:**
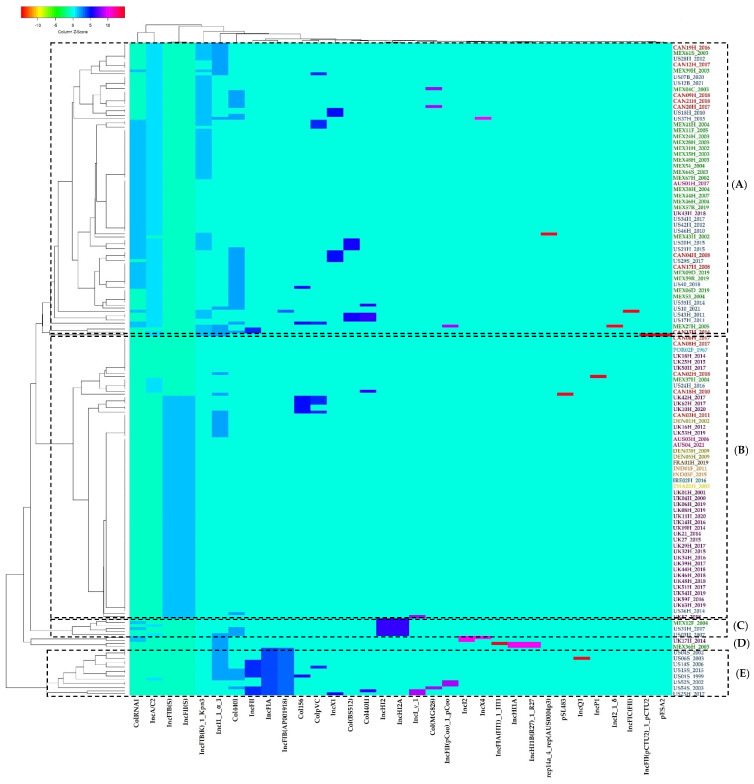
Grouping pattern by strain and plasmid presence/absence in the genomes of the ST213 strains analyzed in this work. A binary matrix of presence (1)/absence (0) of each shown plasmid was used as an input for grouping with the Manhattan and UPGMA methods for the calculation of distances and the generation of the grouping pattern, respectively. See Section 4 for details. (**A**) Grouping pattern of genome/strain from UK and one from Australia; (**B**) Grouping pattern of genome/strain European, Asian, and Australian, with five samples from Canada, one from Mexico, and one from the USA; (**C**) Grouping pattern of genome/strain from Mexico; (**D**) Grouping pattern of genome/strain from the USA; (**E**) Grouping pattern of genome/strain from the UK. Country symbols: **AUS**, Australia; **BEL**, Belgium; **CAN**, Canada; **DEN**, Denmark; **FRA**, France; **IND**, India; **IRE**, Ireland; **MEX**, Mexico; **NET**, Netherlands; **POR**, Portugal; **THA**, Thailand; **UK**, United Kingdom; **US**, United States of America.

**Figure 8 antibiotics-11-00925-f008:**
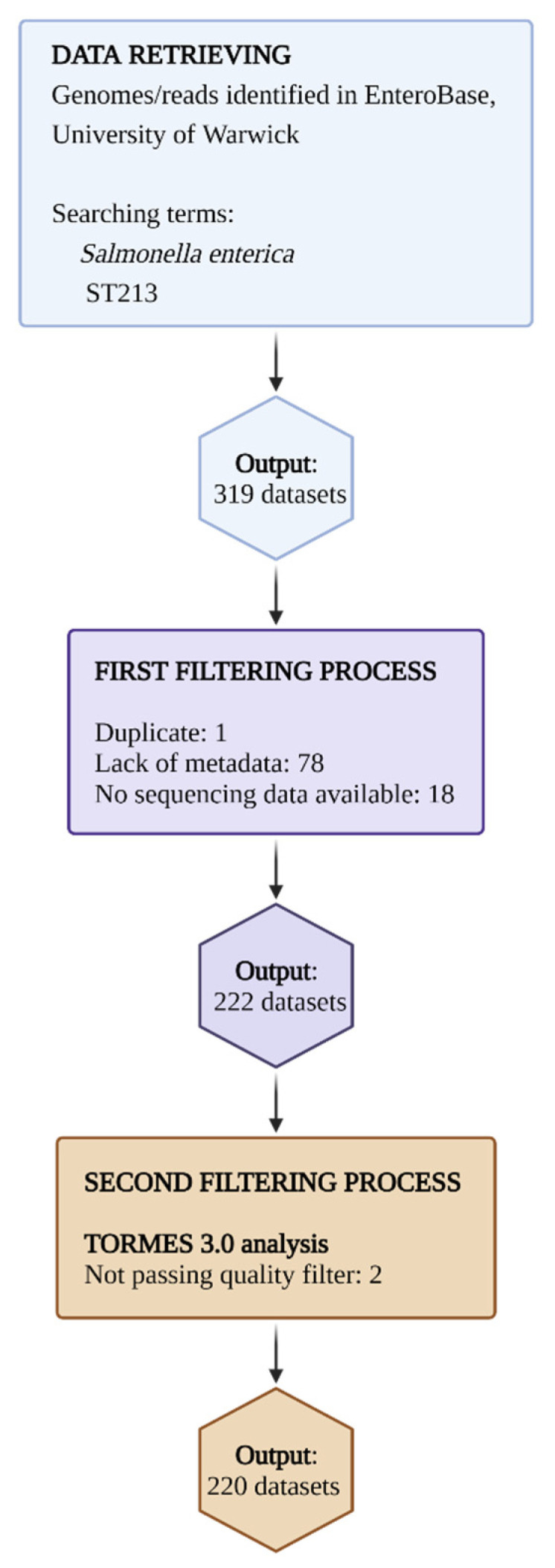
Data retrieval and exclusion criteria used for the analyzed genomes. The flowchart shows the number of genomes of read datasets initially found in EnteroBase and after the filtering criteria were applied. Each filtering criterion shows the number of genomes/reads excluded. The metadata considered to include a genome/reads dataset were country of precedence, type of sample, and year of isolation. For details on the analyzed genomes, see Appendix A. For quality filter analysis, see the main text (figure created with Biorender.com, accessed on 1 July 2022).

**Table 1 antibiotics-11-00925-t001:** Resistance genes found in the analyzed genomes of ST213 strains.

Antibiotic Group/Resistance Gene(s)	Encoded ^1^	% Frequency ^2^
Aminoglycosides		
*aph(4)-Ia*, *aac(3)-IV*, *ant(3)-IIa*, *aac(3)-IId*	P (IncHI2)	0.4 (*n* = 1), 0.9 (*n* = 2), 10.9 (*n* = 24), 29 (*n* = 64)
*aph(3)-IIa*, *aph(3)-Ia*	T	0.9 (*n* = 2), 9.5 (*n* = 21)
*aac(6)-Iaa*	C	100 (*n* = 220)
*aadA2*	P/I (IncHI2/IncHI2A)	40 (*n* = 89)
*aadA5*	P/T/I	1.8 (*n* = 4)
*aph(3)-Ib*	P/T/C (IncC, IncFII/IA/IB,)	50 (*n* = 111)
*aph(6)-Id*	P/CG-I (IncC, IncFII/IA/IB,)	50 (*n* = 110)
Cephamicin		
*bla_CMY-59_*	P (IncC)	36.8 (*n* = 81)
Diaminopyrimidines		
*dfrA1*, *dfrA12*, *dfrA17*	I	0.4 (*n* = 1), 37.2 (*n* = 82), 1.8 (*n* = 4)
Penam		
*bla_CARB-3_*	P	0.4 (*n* = 1)
Penam, penem, cephalosporin, monobactam		
*bla_TEM-1_*	C/P (IncHI2/IncHI2A)	14 (*n* = 31)
Disinfecting agents and intercalating dyes		
*qacH*	P	26.3 (*n* = 58)
Fluoroquinolones		
*mdtK*	C	100 (*n* = 220)
*qnrA1*, *qnrB5*, *qnrB19*, *qnrS1*	P (IncHI2/IncHI2A)	0.4 (*n* = 1), 10 (*n* = 22), 0.4 (*n* = 1), 0.9 (*n* = 2)
Lincosamides		
*linG*	I-agc, with *aadA2*	0.4 (*n* = 1)
Macrolides		
*mefB*	P, located in the *sul3* vicinity	0.4 (*n* = 1)
Sulfonamides		
*sul1*	C-1 I	8.6 (*n* = 19)
*sul2*	SP (IncHI2, IncC, IncFII/IA/IB)	49.5 (*n* = 109)
*sul3*	P (IncHI2/IncHI2A)	27.2 (*n* = 60)
Tetracyclines		
*tetA*, *tetB*, *tetR*	C/P (IncHI2, IncC, IncFII/IA/IB)	56.8 (*n* = 125), 1.8 (*n* = 4), 55.9 (*n* = 123)
*tetC*	P (IncHI2/IncHI2A/IncI1_I_γ)	1.3 (*n* = 3)
*tetU*	P (pKQ10)	0.4 (*n* = 1)
*tetM*	T	0.4 (*n* = 1)
Phenicol		
*catII (E. coli K-12)*, *cmlA1*, *floR*	C/P (IncHI2/ IncHI2A/ IncI1_I_γ/IncQ1, IncC)	0.4 (*n* = 1), 0.9 (*n* = 2), 47.7 (*n* = 105)
Phenicol, β-lactams, diaminopyrimidines, fluoroquinolones, glycyl-cyclines, nitrofuran and tetracyclines, rifamycin, triclosan.		
*golS*, *mdsA*, *mdsB*, *mdsC*	C	100 (*n* = 220)
*oqxA*, *oqxB*	C/P (IncHI2)	24.5 (*n* = 54)
*sdiA*	C/P	100 (*n* = 220)

Note: ^1^ C, chromosome; CG-I, chromosome genomic islands; C-1 I, class 1 integron; I, integrons; Iagc, Integron-associated gene cassette; P, plasmids; SP, small plasmids; T, transposon. ^2^ Number of genomes in which each gen/plasmid was found is given in parentheses.

**Table 2 antibiotics-11-00925-t002:** Mutations associated with antibiotic resistance in the analyzed genomes.

Mutation	Codon Change	Amino Acid Change	Genome	% of Genomes ^1^
*gyrA* p.D87G	GAC → GGC	D → G	MEX04C_2003	3.4 (*n* = 1)
*parC* p.S80I	AGC → ATC	S → I	US37H_2015	3.4 (*n* = 1)
*gyrA* p.S83F	TCC → TTC	S → F	MEX22F_2008	6.8 (*n* = 2)
			MEX51H_2011	
*gyrA* p.D87N	GAC → AAC	D → N	US17H_2011	10.3 (*n* = 3)
			US19H_2011	
			US43H_2011	
*gyrA* p.S83Y	TCC → TAC	S → Y	CAN20H_2017	79.3 (*n* = 23)
			MEX14F_2009	
			MEX15F_2009	
			MEX16F_2009	
			MEX18F_2008	
			MEX34H_2003	
			MEX56R_2019	
			THA01C_2019	
			US07B_2020	
			US12B_2021	
			US18H_2010	
			US20H_2015	
			US21H_2015	
			US22H_2015	
			US24H_2016	
			US28H_2012	
			US30H_2017	
			US37H_2015	
			US38H_2017	
			US44H_2010	
			US46H_2010	
			US47H_2017	
			US49B_2019	

Note: ^1^ Percentage of genomes that carry each plasmid. The number of genomes for each case is shown in parentheses. The total number of analyzed genomes was 220.

## Data Availability

Date is available in the Appendix A.

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
