# Peer review of "Comparative Genomic Analysis Discloses Differential Distribution of Antibiotic Resistance Determinants between Worldwide Strains of the Emergent ST213 Genotype of *Salmonella* Typhimurium"

_antibiotics, 2022, doi:10.3390/antibiotics11070925_

Round 1
Reviewer 1 Report
The study is well carried out and the results are very well justified and discussed.
Apart from minor grammatical changes I would suggest that Figures 1,2 and 3 be converted into a table format.
Discussion can be shortened.
Reviewer 2 Report
In addition to the thorough language check, I have the following suggestions for the authors:
Page 14 Line 247: “Same serotype, ……..” This sentence looks incomplete.
Page 15 Line 299: “…….in humans of strains……” Recheck the sentence.
Page 17 Line 392: “….. only describing…..” → “…… only described…...”
These were some examples. Such issues were detected throughout the manuscript and must be checked thoroughly by the authors.
Page 19 Lines 488-490 and 491- 492: These two statements seem contradictory because in the first sentence the authors mention that 12 plasmids were associated with colistin resistance genes (e.g. mcr). However, in the very next sentence, they state that no mcr gene was identified. This contradiction if any should be made clear.
The discussion is extremely long and it must be trimmed for a better read.
Supplementary Table S1 is not provided so it is difficult to check the final dataset.
What is the status of all the genomes used in this work? How many are drafts, or complete? What is the range of contig numbers in the genomes at the draft level? These statistics should be provided because analyzing genomes with multiple contigs can influence the overall results.
Reviewer 3 Report
The manuscript is of great interest to scientists working on antibiotic resistance and its spread among different countries and isolates. The authors compared the genome sequences of ST213 strains pointing out on similarities and differences among strains. Although the limitations of the present comparison, that the authors highlight at the end in the Final considerations, I consider the manuscript suitable for publication after a deep revision of the English. Listed are some examples:
L27-28 rephrase the sentence: "Antibiotic resistance genes/plasmids groups genomes according with the strain geographic precedence and sample type of isolation."
L28-29 "Genetic determinants of antibiotic resistance groups the genomes of North America" and "Results obtained here highlights" please pay attention to the singular and plural.
L73: avoid repetition, such as "the treatment of HIV treatment"
Along with the text, the authors write "study genomes", do they mean the studied genomes?
L247: "Same serotype, as shown by several recent reviews on the subject [2,8,39,24,40]." A verb is missing.
L274: rephrase the sentence: "Isolation year of ST213 strains whose genomes were here analyzed, suggest its recent dispersion to India and Australia."
L283: correct Ocania.
Furthermore, the discussion is too long and the reader does not clearly read the importance of the results. The authors should try to reduce it and avoid reporting the results.
Overall, the manuscript is clear, but the words in the figures are blurred.
My regards
